# Anatomic Variations Important for Dental Implantation in the Mandible—A Systematic Review

**DOI:** 10.3390/diagnostics15020155

**Published:** 2025-01-11

**Authors:** Zlata Rajkovic Pavlovic, Milos Stepovic, Marija Bubalo, Ivana Zivanovic Macuzic, Maja Vulovic, Nevena Folic, Jovana Milosavljevic, Valentina Opancina, Dobrivoje Stojadinovic

**Affiliations:** 1Department of Dentistry, Faculty of Medical Sciences, University of Kragujevac, 34000 Kragujevac, Serbia; zlatakg@yahoo.com (Z.R.P.);; 2Department of Anatomy, Faculty of Medical Sciences, University of Kragujevac, 34000 Kragujevac, Serbia; 3Department of Pediatrics, Faculty of Medical Sciences, University of Kragujevac, 34000 Kragujevac, Serbia; 4Department of Radiology, Faculty of Medical Sciences, University of Kragujevac, 34000 Kragujevac, Serbia

**Keywords:** mandibular canal, mental foramen, mandibular concavity, incisive canal, lingual canal, implantation, CBCT

## Abstract

**Background**: This is a systematic review on the subject of anatomic landmarks and variations in the mandible that influence implantation placement. With this systematic review, we would like to summarize the results from different studies that are relevant to this subject and that are up to date, presenting their main findings, the measurements of mentioned landmarks, and giving clinical implications that will be helpful to practitioners in their better understanding of this topic. **Methods**: This study followed all of the elements of PRISMA. The criteria for inclusion and exclusion are described in detail. The following bibliographic databases were searched: PubMed (MEDLINE), The Cochrane Library, Wiley Online Library, EMBASE, and, additionally, Google Scholar. The search of articles was carried out using a combination of different keywords with a Boolean operator for each keyword. A total of 30 studies were included in this review and the risk of bias for each study was assessed. This review was registered with the PROSPERO ID number CRD42024609308. **Results**: The structure “SPIDER“ was used to present the findings in the summation table, followed by a detailed description of the quantitative findings and overall mean values of the most commonly used measured points. The morphometric measurements of anatomic details and variations in the mandible, such as the mandibular canal, mental foramen, lingual foramina, lingual canal, incisive canal, and mandibular concavity, are of high significance for clinicians dealing with implantation where gender, ethnicity, age groups, the side of the mandible, or remaining teeth can influence implantation planning. **Conclusions**: The distances of the named anatomic landmarks to the surrounding bone structures that are used as a guide during planning are of huge importance, so proper and detailed measurements must be executed by experienced professionals using CBCT. Knowledge of the position and variation in these landmarks may be used for any bone-guided augmentation, surgical transpositions of anatomic landmarks, and surgery intervention planning. The results of this study can aid in choosing appropriate measurement points and give the gross picture, for clinicians, in therapy planning, considering all the landmarks of significance in the mandible.

## 1. Introduction

The modern era brings the possibility of the rehabilitation of missing teeth using dental implants [1]. The possible modalities of therapy vary for cases in which a single tooth is missing, using implants for supporting fixed prostheses [2] and all-on-four and all-in-six concepts [3], to the possibility of retaining the mobile prosthesis [2]. Implant use is increasing every year, regardless of the high cost, and it is related to high implant survival and success rates (over 95%) [4,5]. Consequently, the number of complications, such as neurosensory trauma and hemorrhage, are rising [6]. This induced the necessity for more detailed analyses of anatomic structures, which can be a danger zone for implant placement, to avoid possible intraoperative and postoperative complications, which can be a significant reason for the lack of success in therapy with consequential implant loss [1].

A 3D reconstruction of the jaw must be provided for planning the ideal spot for either surgical or prosthetic-guided implant insertion [7]. Cone beam computed tomography became the standard procedure in dentistry preoperative diagnostics due to its low cost, lower radiation dose in accordance with ALARA and ALADA standards, low exposure time, great visualization of anatomic structures, quantitative and qualitative bone analyses, and possibility of virtual implant placement and planning through the design of the prosthetic work with implants utilizing software tools [8,9,10].

The guide for implant placement must provide the ideal platform to restore aesthetics and functions, and anatomic limits must be considered, and preoperative planning must be executed with a high level of precision and details [1]. After tooth loss, the residual alveolar ridge is susceptible to the bone remodeling process, which may lead to the transposition of important anatomic structures, like the position of the inferior alveolar canal and inferior alveolar nerve (IAN) [11], or the mental foramen and nerve, that can be repositioned on the top of the residual ridge or in its close relation [1]. Studies showed differences in its vertical position in edentulous patients, but also in the horizontal position of the IAN, leading its trajectory to follow the lingual plate, middle of the buccolingual, or changing direction on its way [12]. The mental foramen, after bone remodeling, could be positioned on top of the residual ridge, making the use and wearing of the total dental prosthesis uncomfortable or impossible [13].

Lately, the anterior part of the mandible, despite opinions of it being a reliable area for implantation, showed that the damage of some anatomic characteristics in the interforaminal region during surgical procedures may be a potential reason for postoperative complications [14]. Also, the literature reports complications related to the laceration of the incisive canal, including postoperative extensive mouth floor hematoma, neurosensory disorders (like hypoesthesia of the lip and pain in the anterior teeth), and aggravation of the implant surgical procedure [15]. A variety of lingual foramina, including their shape, trajectory, and number of canals, are described in many studies in dentulous patients, but in edentulous jaws, they can stay undetected, causing severe bleeding by canal interruption during implant bed preparation [16].

The morphology of the mandible is defined to be in close proximity to anatomic features, like the sublingual and submandibular salivary glands located in their respective fosses or mylohyoid line for muscle attachment, which may influence dimension of the bone, especially after tooth extraction [17]. There is an undeniable importance of anatomic concavity of the lingual part of the mandible, which after the remodeling of the residual ridge may mislead the implant position causing perforation of the lingual cortex and trauma of nearby anatomic structures [18].

With this systematic review, we would like to analyze and discuss the influence of anatomic details of the mandible on the dental implantation placement, which must be considered prior to implant planning. Adequate radiological planning and determination of the precise position of the dental implants using CBCT (cone beam computed tomography) are crucial for successful therapy. In this review, we will present the findings of studies from the past ten years that focus on the following:Influence of the mandibular canal and its morphometric characteristics on the dental implant placement;Influence of the mental foramen and its morphometric characteristics on the dental implant placement;Influence of the incisive canal of the mandible and its morphometric characteristics on the dental implant placement;Influence of lingual foramina and lingual canals and their morphometric characteristics on the dental implant placement;Influence of mandibular concavity and its morphometric characteristics on the dental implant placement.

## 2. Materials and Methods

This study followed all the elements of PRISMA (Preferred Reporting Items for Systematic Reviews and Meta-Analysis) [19]. This review was registered with PROSPERO ID number CRD42024609308.

### 2.1. Eligibility Criteria

Only articles with free full-text availability, conducted in the last ten years (2014 to 2024), in English, on humans, dealing with measures of specific anatomic landmarks in the mandible, and being necessary for implantation planning were included. Review articles, case reports, pilot studies, unpublished manuscripts, and conference abstracts were omitted. If the analysis of anatomic details that we considered was not measured or not reported, those studies were considered ineligible.

### 2.2. Information Source

All databases were searched between 9 October and 1 November 2024. The following bibliographic databases were searched: PubMed (MEDLINE), The Cochrane Library, Wiley Online Library, EMBASE, and, additionally, Google Scholar, in the period of one decade. 

### 2.3. Search Strategy and Selection Process

Two reviewers (ZRP and MS) independently screened each record, by title and abstracts, and retrieved each report, and afterward duplicated studies were removed from further analysis. Where necessary, a third reviewer (JM or VO) was consulted to resolve disagreements between screeners. The search of the articles was carried out using a combination of different keywords with a Boolean operator for each word combination. The following keyword combination was used: implantation AND mandibular canal AND CBCT, implantation AND mental foramen AND CBCT, implantation AND lingual foramen AND CBCT, implantation AND mandibular incisive canal AND CBCT, implantation AND mandibular concavity AND CBCT.

The number of articles found in each database, like the process of removing duplicates and the selection of articles, is presented in the PRISMA flow diagram (Figure 1) [20]. The total number of articles included in this systematic review was 30.

### 2.4. Study Eligibility and Quality Assessment

After inclusion and inclusion criteria defined with search strategies of databases, the articles were then critically considered based on their abstract and their relevance for the subject, and finally, articles had to pass the eligibility criteria performed by ZRP, MS, JM, and VO, before being included in the final number.

Study eligibility was assessed by a checklist for analytical cross-sectional studies [21]. This checklist consists of 8 questions with a particular view on what counts as evidence and the methods utilized to synthesize those different types of evidence with critical appraisal and synthesis of these diverse forms of evidence to aid in clinical decision-making in healthcare. Each question had 4 possible answers: yes, no, not sure, and not applicable. Articles where more than half of the answers were positive were described as eligible by their quality. The following questions were set for an answer:Were the criteria for inclusion in the sample clearly defined?Were the study subjects and the setting described in detail?Was the exposure measured validly and reliably?Were objective, standard criteria used for measurement of the condition?Were confounding factors identified?Were strategies to deal with confounding factors stated?Were the outcomes measured validly and reliably?Was appropriate statistical analysis used?

### 2.5. Data Items

#### 2.5.1. Outcomes of Interest

Outcomes of interest for our systematic review were measurements of anatomic landmarks/variations—mandibular canal, mental foramen, lingual foramen, incisive foramen, and canal and mandibular concavity. Those measurements help in guiding the implantation placement in the anterior and the posterior part of the mandible, which is crucial for successful and safe implantation. Determined points for measurements of these landmarks are analyzed and described, and only articles using CBCT as the method for analyzing the implant placement are considered, as this method is the golden standard for implant placement.

#### 2.5.2. Population

CBCT scans were taken in specialized dental centers with a 3D CBCT machine and visualized in CBCT software. Inclusion criteria were patients over 18 years old and permanent dentition, absence of serious pathological lesions in the mandible, absence of previous trauma, surgical interventions, congenital syndromes, fractures or foreign bodies (which produce artifacts in scans), good quality of CBCT images of the mandible, absence of severe atrophy of the mandible, and absence of any impacted teeth. Depending on the variety of studies, partially edentulous, totally edentulous patients, or both were included. The exclusion criteria depend on the region of interest observed such as severe or progressive periodontitis, abnormal tooth development, apical cysts, intra- and extra-root resorption, history of dental and jaw trauma, history of orthodontic treatment, and low-quality CBCTs.

#### 2.5.3. Intervention

Presurgical planning of implant placement considers morphological analyses of anatomy structures that can be of decisive importance for the long-term success of implant therapy. Also, anatomic visualization is important to avoid possible intraoperative and postoperative complications like hemorrhage, edema, paresthesia, and early implant loss. The golden standard for the visualization of the bone structures of the upper and lower jaw is CBCT (cone beam computed tomography). CBCT has become increasingly important in planning dental implant placement and CBCT scanners are now finding many uses in other fields of dentistry as well, such as oral surgery, endodontics, and orthodontics. During dental/orthodontic imaging, the CBCT scanner rotates around the patient’s head, obtaining up to nearly 600 distinct images. This give scans of excellent precision in all planes necessary for analysis before implant placement

### 2.6. Effect Measures

The main outcome of the effect measures important for our study were distances of the anatomic landmarks in the mandible from their specific points relevant to the implantation process, expressed in the mean or median values, with standard deviations. Distances of anatomic landmarks are measured in millimeters in the specific programs for CBCT scan analysis.

Additional outcomes are a variation in measurements according to the sex and age groups and different groups of teeth (dentate, edentulous, or partially edentulous) or side of the jaw (left and right), also expressed as the mean or median values in millimeters, where available, and significant relationship expressed as *p* values between mentioned measured anatomic details.

### 2.7. Risk of Bias of Included Articles

The risk of bias was measured with the ROBINS-I V2 tool that considered the following: Domain 1: risk of bias due to confounding, Domain 2: risk of bias in classification of interventions, Domain 3: risk of bias in selection of participants into the study (or into the analysis), Domain 4: risk of bias due to deviations from intended interventions, Domain 5: risk of bias due to missing data, Domain 6: risk of bias arising from measurement of the outcome, Domain 7: risk of bias in selection of the reported result. Questions have response options yes, probably yes, probably no, no, and no information. Judgment based on the answers can be a low, moderate, serious, and critical risk of bias [22,23].

### 2.8. Synthesis Method

This systematic article focuses on the qualitative data synthesis and quantitative presentation of the main outcomes and findings of included studies that will be presented in the results using the SPIDER (Sample, Phenomenon of Interest, Design, Evaluation, Research type) method with few additional labels specific for this study [24].

## 3. Results

### 3.1. Study Characteristics

The main characteristics of included studies are presented by the SPIDER method in Table 1, including author name, year of the study, country, sample size, age, the phenomenon of interest, design, evaluation, and research type. Overall means of the most commonly measured anatomical point for each observed landmark are presented in the tables below their corresponding subheadings in the results. 

Most of the studies—24 out of 30—had a low risk of bias. The remaining six studies had some concerns, mostly considering Domain 3 because either the inclusion and exclusion criteria were not clearly stated or the sample size was questionable due to the lack of an explanation. Followed by Domain 7, because of missing certain tables, not the all described results were visible, and Domain 2 because the CBCT unit/software was not stated. Details can be seen in Table 2.

### 3.2. Results of Individual Studies

#### 3.2.1. Inferior Alveolar Canal (IAC)

Aljarbou et al. (2019) [25] determined that the distance between root apices of the mandibular molar and the inferior alveolar canal (IAN) ranged from 1.68 to 4.79 mm. The second molar apices were closer to the IAC, but considering the mesial and distal roots of a first and second permanent molar, the distal roots were closer to the IAN. Considering gender, the females had a significantly closer relation between the IAC and distal root of second molars. Still, no differences were found considering the left and right sides of the mandible.

Koivisto T et al. (2016) [38] aimed to find the location of the mandibular canal in its relation to the apices of mandibular teeth where position varied and it was located inferiorly, more towards the buccal side, or posteriorly, to the lingual side. Its position was the most common on the buccal side when observing the second molar and inferior to the first molar and second premolar. Buccal bone thickness over the mandibular canal was thickest at the vertical level of the mesial root of the second molars (mean 5.4 mm), and thinnest at the vertical level of the second premolars (2.6 mm). The lingual bone over the MC was thickest at the second premolars (3.8 mm) and thinnest over the distal root of the first molar (1.7 mm). They also measured the average diameter of the mandibular canal from the second molar to the second premolar, and it was 3.03 mm on the left side and 2.91 mm on the right side.

The position of the mandibular canal relative to the roots of the mandibular third molar can be apical (88.1%), on the buccal or lingual side (7.9% and 3.5%), or between the roots (0.5%), and only 7.3% of all analyzed third molars had a direct relation with the mandibular canal as showed by the Gu L et al. (2018) [53]. If there was contact, it was significantly proven that the root tends to be on the lingual side, and on the contrary, it was located apical if there was no contact.

Kavarthapu A et al. (2018) [46] presented the average parameters of the position and course of the IAN from visualized borders in the mandible, that are important for implantation. The average value of the distance from the alveolar crest to the superior border of the IAN varied from 14.75 ±3.02 mm to 15.38 ± 5.37 mm depending on the observed section, and in Section 1 the IAN was more inferiorly positioned in males, but no age difference was found, nor the difference between the partially or totally edentulous participants. The difference was also found in the distance from the lingual cortex in the dentulous group where the IAN was further away, and the distance from the inferior border of the mandible to the inferior border of MC where the IAN was positioned higher in the dentate group at Sections 2 and 5.

In the results of Genç T et al. (2018) [29], the mean vertical size of the mandibular canal was 2.71 ± 0.52 mm in males and 2.34 ± 0.50 mm, and significance was found between genders for the mean measurements of the distance between the mandibular canal and crest (male, 12.46 ± 3.17; female, 10.69 ± 3.67) and mandibular canal and inferior border of the mandible (male, 8.05 ± 1.51; female, 7.11 ± 1.14).

Shen YW et al. (2021) [31] investigated the prevalence of anatomic variation in the retromolar canal, which typically originates in the mandibular canal on the distal side of the third molar and extends to the retromolar foramen (RMF). They found that the prevalence of this foramen was in 10% of males and 10.5% of females, where gender and the side of the mandible had no correlations for its appearance. The average diameter of the RMF was 1.41 ± 0.30 mm, the height was 13.62 ± 1.34 mm, and the horizontal distance from the retromolar canal to the second molar was 11.57 ± 2.70 mm. Overall means from the most common anatomical measured points for the inferior alveolar canal can be seen in Table 3.

#### 3.2.2. Mental Foramen (MF) and Anterior Loop (AL)

Genç T et al. (2018) [29] confirmed the location of the mental foramen, and it was mostly present between the first and second permanent premolars. On average, 12.92 mm of alveolar bone was found to be coronal to the mental foramen presented by Raju et al. (2019) [44]. The mean distance from the lower point of the mental foramen to the lower point of the mandible was 11.4 mm, and the mean distance from the root of the dental canal to the upper point of the mental foramen was 2.17 mm as shown by Vujanovic-Eskenazi. The diameter of the mental foramen was larger in males (3.27 ± 0.78 mm in males and 2.87 ± 0.64 mm in females), as well as the distance from the mental foramen to the inferior border of the mandible (males, 12.13 ± 2.01 mm; females, 10.87 ± 1.71 mm). The vertical mental foramen diameter was 3.31 ± 1.01 mm and the horizontal MF was 2.80 ± 0.99 mm according to the results of Gümüşok M et al. (2017) [35].

Both studies (J, PC et al. (2018) [42] and Raju et al. (2018) [44]) which described the occurrence of anterior loops found them to be more bilaterally (60% and 90%) than unilaterally (40% and 10%). The percentage distribution of the anterior loop in the buccal, middle, and lingual one-third of the alveolar ridge was 83.1%, 16.9%, and 0% as demonstrated by Raju et al. [44]. The prevalence of the anterior loop was very variable. Genç T et al. (2018) [29] and Marimuthu et al. (2018) [42] showed the least prevalence, from 5.76 to 11.76%; Raju et al. (2019) [44] showed 25% prevalence; while Wong et al. (2018) [36] and Lu Ci et al. (2015) [43] presented the highest prevalence of 85.2–94% where gender mostly did not have an impact on presence.

On average, the alveolar bone found to be coronal to the anterior loop was 17.06 mm, and the anterior loop was located more apically relative to the mental foramen as presented by Raju et al. [44]. The mean loop length varied too, from 1.46 ± 1.25 mm, 1.59 ± 0.9, 1.63 mm, 2.79 mm, to 3.77 ± 1.74 mm (Lu Ci et al. [43], Vujanovic-Eskenazi A et al. [45], Raju et al. [44], Marimuthu et al. [42], Wong et al. [36]), while the mean length of the anterior loop was 2.53 ± 1.27 mm, and the caudal loop was 6.04 ± 1.66 mm in the study of Yang et al. [39]. Loop length did not differ among gender, sides of the jaw, or dentate status, and Wong did not find a difference among ethnicity groups, and age groups had an influence. Only in the study of Lu Ci et al. [43] was difference between age groups found, where the 21–40-year group (1.89 ± 1.35 mm) had larger mean values compared to those of the 41–60-year group and 61–80-year group. The mean length and diameter of the anterior loop at the origin of the mandibular incisive canal were 9.97 ± 5.15 mm and 1.97 ± 0.48 mm as presented by Yang et al. [39].

Gümüşok M et al. (2017) [35] described anatomic variation in the accessory mental foramen, and nearly 12% of subjects had it in the posterior inferior region. The mean distance between the mental foramen and accessory mental foramen was 2.84 ± 2.14 mm. AMF diameters were as follows—vertically it was 1.50 ± 0.63 and horizontally it was 1.27 ± 0.40. No difference between genders was found. Overall, means from the most common anatomical measured points for the mental foramen and alveolar loop can be seen in Table 4.

#### 3.2.3. Lingual Foramina/Lingual Canal (LF/LC)

Fouda et al. (2020) [28] found that the mean interforaminal distance in edentulous mandibles was 45.77 ± 5.78 mm, significantly less than in dentate mandibles which was 50.44 ± 7.06 mm. According to the results of different research included in this review, the number of foramina varies from none to eight, but the most common number is two, as demonstrated by the results of Sekerci (2014) [26] and Sener (2018) [37], or three to four, as presented by He (2016) [47]. The most common location of lingual foramina was in the center of the symphysis, but looking at the more detailed position by He et al. [47], they were mostly located in the region of the central incisor. The direction of lingual foramina was inclined in most of the female and male respondents in the study by He et al. [47], and foramina in the middle were predominantly located below toot apices. The location of the foramina was also determined by the genial tubercle, and they were divided into the foramina above and below it. Morphometric analyses significantly differed for foramina above tubercle where the vertical distance from the superior border of the mandible to foramina was 12.04 mm and that from the inferior border to the lingual foramina was 18.36 mm, and the foramina found below had a vertical distance from the superior border of the mandible to foramina that was 24.46 mm, and the inferior border to the lingual foramina was 4.94 mm.

The distance from the lingual foramen to the alveolar crest was higher in males than in females (14.70 ± 4.81 mm vs. 12.54 ± 3.89 mm) as shown by the results of Genc et al. [29]. The mean distance between the superior lingual foramen and the base of the mandible in males was 15.53 ± 1.88 mm and in females 14.19 ± 1.66 mm as presented by the results of Surathu et al. [32], and the distance was significantly different according to gender but age did not influence it. The diameter of lingual foramina was mostly less than 1 mm, as demonstrated by Sekerci and He, in both male and female patients.

Alqutaibi found that most of the patients had two lingual canals, while Sanchez-Perez mostly found three. Alquitabi presented that the most common type (95%) of lingual canal was the supra-spinosum canal. The length of the supra-spinosum canal was 5.81 ± 2.08 mm. Supra-spinous measurements regarding the inferior border to buccal and lingual terminal distance were 11.03 ± 2.4 mm and 14.95 ± 2.19 mm, and they did differ by gender. Fouda and Sanches Peres presented the median length of the lingual canal for the dentate and edentulous groups as follows: 8.28 mm in the dentate group and 8.15 mm in the edentulous one, while being 9.25 mm in the dentate group and the 8.60 mm in the edentulous one. The distance from the base of the lingual canal to the alveolar crest was significantly higher in the dentate group compared to in the edentulous group (20.79± 5.84 mm vs. 15.00 ± 4.11 mm), as shown by Fouda. In the study by Sanchez et al. [33], the parameters to be considered when comparing the distance of the lingual canal to the surrounding structures in dentate and edentulous patients were distances to the buccal cortex (4.91 ± 1.38 mm vs. 4.64 ± 1.81 mm), inferior cortex (8.48 ± 2.75 mm vs. 7.01 ± 2.26 mm), and lingual cortex (8.75 ± 0.86 mm vs. 8.48 ± 1.86 mm). The diameter of the canal in dentate patients was 1.39 ± 0.46 mm and in edentulous patients was 1.48 ± 0.45 mm. Overall means from the most common anatomical measured points for lingual foramina and the lingual canal can be seen in Table 5.

#### 3.2.4. Mandibular Incisive Canal (MIC)

Yang et al. (2018) [39] presented that the mandibular incisive canal mostly ends up between the lateral incisor and canine and that the diameter was longer in males than in females (2.05 ± 0.48 vs. 1.92 ± 0.47 mm) while the length of the mandibular incisive canal increased with tooth loss (completely edentulous 14.31 ± 1.83 mm, partially edentulous 10.91 ± 5.20 mm, and dentate 9.41± 5.05 mm).

Barbosa et al. (2020) [14] showed the mean length of the mandibular incisive canal was 7.70 ± 3.70 mm and did not differ between sides. Sener et al. (2018) [37] showed that the mean incisive length was significantly different between the edentulous (3.08 ± 1.70 mm) and dentate group (2.55 ± 0.81 mm). The mean diameter of the mandibular incisive canal in the dentate groups was 2.44 ± 0.702 mm and in the edentulous groups was 2.35 ± 0.652 mm.

Barbosa also showed that linear measurements from the MIC to the alveolar cortex (mean, 16.63 ±7.98 mm) and to the buccal cortex (mean, 2.60 ± 1.37 mm) for the right side of the mandible were significantly greater compared with those for the left side. Males had greater mean length values at their initial or final portions to the alveolar crest (17.23 mm and 19.55 mm), buccal cortex (2.75 mm and 4.32 mm), and inferior border of the mandible (9.98 mm and 9.60 mm) than females. Mean distances from the MIC to the adjacent teeth apexes ranged from 6.97 ± 1.58 mm to 9.77 ± 1.04 mm, and no statistically significant difference was found. Overall means from the most common anatomical measured points for the mandibular incisive canal can be seen in Table 6.

#### 3.2.5. Mandibular Concavity

Mandibular concavity can be observed anteriorly and posteriorly. The anatomic detail that can influence the implant position posteriorly is the submandibular fossa (SF). Bayrak et al. (2018) [27] investigated the depth of the SF, and the average depth was 2.85 ± 0.8 mm with significant difference between sides (the right side had a bigger depth) and without gender difference. The buccal cortex to mandibular canal thickness (mean 5.02 ± 1.32 mm) was significantly larger in people aged less than 20 years, males, and on the right side, the lingual cortex to the mandibular canal (mean 1.4 ± 0.85 mm) was larger only according to the side, and it was thicker on the left side, and the inferior cortical border to MC (6.7 ± 2.53 mm) was thicker in males older than 45 year.

Vasegh et al. (2022) [34] presented the mean ridge height in the anterior mandible (19.32 ± 6.02 mm) and posterior mandible (12.12 ± 4.81 mm), with a statistically significant difference. The mean ridge width in the anterior mandible (4.48 ± 1.11 mm) compared to the posterior one (5.23 ± 1.17 mm) was also significant.

Zhang et al. (2015) [41] found significance in the posterior ridge heights considering gender, and males had higher mean values than females (26.8 ± 3.3 mm vs. 24.0 ± 3.2 mm), and they also had larger alveolar width (coronal, middle, apical width) than females. The height of the alveolar ridge was greater in the dentate than the edentulous regions, with a significance reached at the first molar and regions of the second and third molar. The dentate ridge also demonstrated a significantly larger alveolar width compared to the edentulous regions, especially for coronal and middle width. Generally, the edentulous ridge had a decreased buccolingual dimension of 2.7 mm (coronal third), 1.0 mm (middle third), and 0.4 mm (apical third). Alqutaibi et al. (2024) [30] investigated posterior dimensions considering age and found that in the age group older than 30 years the height of the ridge was significantly higher (11.45 ± 2.73 mm vs. 10.58 ± 2.61 mm), while the crest width was significantly lower (13.45 ± 2.52 mm vs. 14.53 ± 2.67 mm). The U ridge was more frequent in the region of the second molar compared to that of the first molar and it was more common among males.

Kong et al. (2021) [49] investigated the position of teeth in alveolar processes to the basal bone, which can influence the concavity. The most frequent position was a straight type (basal bone and alveolar process were nearly aligned), an oblique type (the alveolar process was buccally angled with the basal bone), and a concave type (the alveolar process was lingually angled with the basal bone).

Çitir et al. (2021) [48] considered the anterior mandible and concluded that the most common shape was type II (inclined toward lingual). Type I (lingual concavity) was more often found in females and type II in males, while type III (enlarging toward lingual) was the most common type considering both genders, while the least common one had buccal concavity (type IV). Considering bone height and width, the maximum bone height in males was 18.65 to 37.32 mm and in females 13.29 to 32.92 mm. The maximum bone width in males ranged from 9.33 to 16.31 mm and in females from 8.60 to 18.47 mm. The study showed a statistically greater height of bone in males than in females and in the dentate compared to in the edentulous patients.

Vasegh et al. (2022) [34] presented the mean buccal and lingual concavity for the anterior mandible (1.74 ± 0.67 mm and 2.13 ± 0.48 mm) and for the posterior mandible (1.21 ± 0.48 mm and 1.57 ± 0.55 mm). The buccal concavity was significantly different in both the anterior and posterior regions.

Yoon et al. (2017) [40] specified that the posterior mandible was concave in most patients, followed by being parallel, while the anterior region was more parallel and convex. The average concavity length in the anterior region was 18.66 mm in males and 17.80 mm in females. In the posterior right region, the average length was 16.20 mm in males and 15.78 mm in females, and in the posterior left region, the average length was 16.27 mm in males and 16.14 mm in females. According to gender, the degree of posterior concavity varied around 75 degrees in both males and females while the degree of lingual concavity anteriorly was 83.65 in males and 82.88 in females, which differs from Çitir et al. [48] where it was 76.5 ± 3.69.

Tsai et al. (2021) [50] analyzed the crestal and radicular dentoalveolar bone of mandibular anterior teeth, and based on the used categorization, most subjects were in class IV (the crestal and radicular bone was thin). Among class IV, canines had the highest prevalence, and the potential risk for labial bone perforation (LBP) was highest in the canine region (21.6%). Class II (with the crestal bone being thick and radicular bone being thin) had the highest probability of LBP. Concavity depth and torque were statistically higher in the group of patients with perforation while the concavity angle was significantly lower too. Kanewoff et al. (2024) [52] investigated the frequency of perforation of the cortical bone, and perforation was higher in implants planned in the prosthetic-driven position compared to in the bone-driven implant position. Results showed a statistically significant difference in the mean labial concavity angle considering all anterior teeth between males and females. Also, the mean values of mandible basal bone height, mandibular bone thickness at the tooth apex, being 4 mm inferior to the apex were significantly higher in males. Overall means from the most common anatomical measured points for anterior and posterior mandibular concavity can be seen in Table 7.

## 4. Discussion

Using CBCT developed new strategies for finding, classifying, and measuring the morphometric parameters of anatomic structures [54,55], and it may be a key factor for therapy success. Furthermore, it can be useful for predicting the possibility of postoperative complications and their prevention. In our investigation, the most used CBCT scans were the Kodak 9500 CB 3D machine and software [28,37,41,44,45,47], i-CAT 3D imaging and i-Cat Vision software [14,27,38,43,52], Kavo 3D eXam/Kavo Radiant [30,36,49,51], and Planmeca Promax 3D [25,33,35,53], New Tom VGi/NNT 3D [34,39,50] and Sirona XG-3 GAlaxix/Galileo [40,46,48]. The use of CBCT is essential in planning implant therapy in the mandible, according to the variety of anatomic structures that must be considered during deciding implant placement.

The IAN is the final, strongest, mixed branch of the mandibular nerve, which passes down from the infratemporal fossa, leads between the internal and external pterygoid muscle and gets to the ramus, and enters through the mandibular foramen into the mandible canal [56]. The proximity of the canal to mandibular posterior teeth and residual bone after teeth loss presents the IAN as an important anatomic structure in oral surgery planning and implantology interventions [1]. Radiographically, the localization of the mandibular canal can be classified as follows: high, within 2 mm of the apices of the first and second molars; mediate; low; and other variations considering the duplication or division of the canal, partial or complete absence of the canal, or a lack of symmetry [11]. Studies showed various distances from the residual bone to the mandibular canal, and according to the results, they are approximately 14.4 mm in edentulous patients [57]. According to the classification of the mandibular canal in the horizontal plane, the canal can follow the lingual cortical plate of the body and ramus (70%), be in the middle behind the second molar, be near the lingual plate through the second and first molar (15%), and be in the middle or the lingual one-third of the ramus and the body (15%) [12]. All those circumstances of anatomic variation in the IAN position may lead to procedures such as ridge augmentation, bone grafts, and the transposition of structures of the mandible canal, to provide the safe zone for implant placement [1]. In our current review and similar research considering this topic, we determined that the region of the second molar, especially the distal root, is the point where the mandibular canal is the closest to the IAN [58].

A severe problem for edentulous mandibles is the poor quality and quantity of residual jawbone, especially in the posterior region, so an atrophic mandible disables wearing removable dentures. In the posterior region, bone loss may lead to an exposure of the alveolar nerve, which can cause pain and discomfort for those patients [59]. After the tooth extraction bundle bone is replaced by a woven one, this leads to a high reduction in height, especially in the area of the buccal plate. The reason for this rapid bone loss is that the buccal plate is thinner, especially in the anterior and premolar regions [60]. Bone thickness is a determining factor of the primary stability of implants, so the thickness of bone is necessary to be examined, and it was observed that the mandible had an increasing bone thickness towards the posterior side [61]. Age and gender did not have an influence in previous studies, but it was observed that the buccal cortex was thinner in females [62,63].

In the region of premolars, the IAN divides into two terminal branches: the mental and incisive nerve [64]. The incisive nerve goes back to the incisive canal while mental branches sometimes can extend backward and upward forming the anterior loop during its course [39]. Distortion in two-dimensional orthopantomography images may be deceived in premolar regions resulting in misleading the assessment of this anatomic structure, so CBCT is the best option for providing reliable data [65].

Strategically important regions for implantation are the mental foramen, its location, the anterior loop and numbers of mental foramina, as well as the interforaminal region which is necessary to be extensively analyzed because the injury of its content is a major complication [66]. The mental foramen was located mostly between the two premolars and the diameter was bigger in males and the anterior loop was longer in males too, with large diversity in its lengths overall, from 1 to 8 mm [67,68]. This is in line with the results of a study included in this review, where only the length of the loop varied from 1.5 to 3.8 mm. It is recommended that a 2 mm safety zone must be present between the implant and coronary aspect of the mental foramen, and the opening angle of the mental foramen must also be carefully observed [69]. Knowing the morphometrics of the mental foramen can influence the diameter and length of the implant.

The region of the symphysis is still considered to be a relatively safe zone for implants. However, some important anatomic structures such as the incisive canal, lingual foramina, and mandibular concavity can influence the planning of surgical procedures. The lingual foramen presents a small opening in the lingual side of the body of the mandible. Radiographically, they can be localized below the apices of the anterior teeth, and they are variable in their size, diameter, length, direction, and localization [13]. According to the localization, lingual foramina can be divided as ones near the midline or ones occurring laterally. The median lingual canal contains the branches of the lingual and sublingual artery, as well as the mylohyoid nerve, while the lateral lingual canal contains branches that originate from a submental and inferior alveolar artery, as well as the inferior alveolar nerve. The injuries of the lingual canal cause neurosensitive disturbance and hemorrhage [70]. Following its trajectory, a lingual canal can be classified as vertical, horizontal, and inclined [47]. The previous studies described the diameter of the lingual canal as ≤1 or ≥1 mm to determine the risk of severe hemorrhage. Results showed that 75.6% of canals were ≤1 mm, with normal disturbance according to gender [71], while others described that the majority (64%) of the medial lingual canals had a diameter ≥ 1 mm, while only 32% of accessory canals had a diameter less than 1 mm [72]. The medial lingual foramen was classified as the supraspinatus and infraspinatus [51], according its relationship to the genial tubercle. In a systemic review [13], we found a higher prevalence of the supraspinatus lingual foramen (over 90%). Studies showed that during dental implant planning more consideration about the median lingual foramen should be given to the elderly and male patients [14].

Postoperative complications after procedures in the anterior mandible were highly presented due to damage of the content of the incisive canal because of its unpredictable position, which was also linked to the failure of successful implant placement [73]. The MIC was not influenced by gender or other sociodemographic factors, but female gender had slightly higher prevalence and bilateral occurrence. Female gender was also confirmed as the riskier gender by this systematic review, but the male mean length was higher in the initial or final portions. The diameter ranged from 0.5 to 2.5 mm and the length from 6 to 18 mm, and the sides of the mandible did not have a significant influence on those measurements. The MIC was closer to the buccal cortical region than the lingual cortical region, and it was up to 20 mm below the alveolar ridge [74].

In the mandible, this buccal plate reduction may lead to unappropriated bone analyses in the posterior region, especially on an orthopantomography view, so lingual concavity may mislead implant planning, presenting enough wide bone in the coronal parts of the residual ridge and not enough bone in the apical part because of anatomic concavity. Also, these boundaries can cause implant placement in a position that can be difficult for prosthetic load [75].

Mandibular concavity is also one of the important landmarks to be considered when planning implant therapy. Due to the different thicknesses of the buccal and lingual cortical plates, and the different angulation of the concavities in the anterior and posterior region due to the existence of anatomic details, certain parts of the mandible must be carefully analyzed to avoid perforation of the cortex when positioning the implant [76]. The alveolar ridge can be classified as convex, parallel, or undercut. Regarding the ridge type, the concavity length, depth, and angle differed in the region of different teeth [77]. Also, parts of the mandible that do not have adequate width can be augmented before the implantation procedure.

All the mentioned elements should be considered and only precise measurements can give relevant information. Due to that, the only acceptable way of measuring anatomic landmarks is cone beam computed tomography. It is considered the golden standard and is acceptable to be used only prior to planning big oral and surgical interventions and implantation. The variation in CBCT performances is related to radiation doses and image quality emphasizes the need for more research to establish proper solutions for three-dimensional imaging following the ALADA principle while focusing on eliminating the sources of artifacts [78].

### 4.1. Limitations of the Evidence Included in This Review

The limitations of the evidence used in this systematic research are inconsistent types of measurements of anatomic details because the comparisons of studies were challenging. We wanted to highlight the variability in the measured anatomical points because measurements on different levels can be an obstacle for future researchers and clinicians in choosing an appropriate point for their evaluation. Finding differences between certain socio-demographic factors, sides, and dentate or edentulous patients was also inconsistent among same anatomic details in different studies because the aims of the study were not the same. Age limit is also important to consider, and because of the radiation and principles of protecting patients, the anatomic details from a younger population are limited.

### 4.2. Implications of the Results for Practice, Policy, and Future Research

The results of this systematic research aimed to combine the information obtained on CBCT scans for the most important anatomic structures in the mandible, which may impact the success of implant rehabilitation. Knowledge of the position and variation may be used for any bone surgery intervention planning. The results from this systematic review can be helpful for future researchers and aid in choosing appropriate measuring points. Furthermore, it can give the gross picture for clinicians in therapy planning, considering all landmarks of significance in the mandible. With this knowledge, the complications of different procedures can be prevented and the therapy outcome can be even more successful. Following the parameter data of the diameter of the canal must be considered due to the possibility of larger vascular branches that can be involved in severe intraoperative and postoperative hemorrhage. Being familiar with the dimensions of the analyzed landmarks is important for planning and controlling intraoperative hemostasis and planning vessel ligature to avoid complications. Standardizing the measurement points for each landmark can help increase the comparability of results from different countries worldwide, which can be the next step for clinicians and researchers. Also, a difference in the results according to gender, age, ethical group, and dentulous/edentulous groups must be considered in further investigation. The obtained results showed limitations in both the anterior and posterior segments of the mandible, yet there is still no clear guide on all the anatomic considerations that must be taken into planning complex implant therapy for the rehabilitation of a segmented or full arch. A further study of CBCT-guided virtual implantation following the observed structures in our research can be of great significance.

## Figures and Tables

**Figure 1 diagnostics-15-00155-f001:**
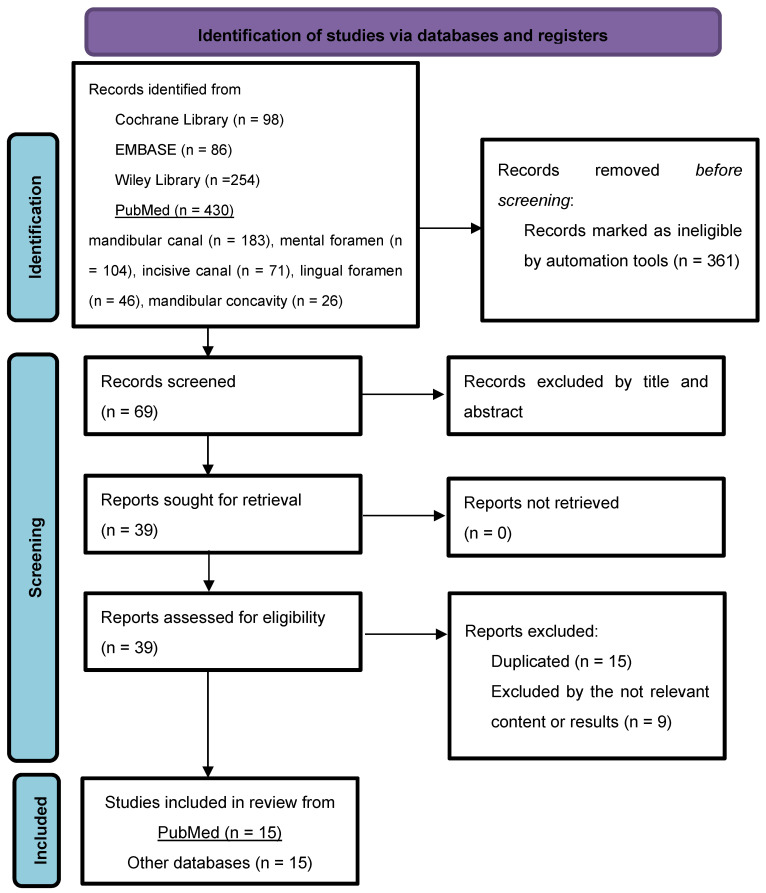
Flow diagram: presenting the number of studies found in each database and detailed search strategy for PubMed.

**Table 1 diagnostics-15-00155-t001:** Main characteristics of included studies: author name, year of the study, country, sample size, age, phenomenon of interest, design, evaluation, and research type.

Study	Year	Country	Sample	Age	Phenomenon of Interest	Design—CBCT Unit/Software	Evaluation	Research Type
Aljarbou FA et al. [25]	2019	Saudi Arabia	60 (36 males, 24 females)	29.59 ± 14.08	Inferior alveolar canal	Cross-section Planmeca Promax^®^ 3D Max/Planmeca Romexis^®^ 3.6	The distance from the outer boundary of the buccal cortical plate to the buccal root surface; the distance from the outer boundary of the lingual cortical plate to the lingual root surface; cortical boundary of the IAC to the nearest root surface.	Quantitative
Sekerci AE et al. [26]	2014	Turkey	500 (237 females and 263 males)	30.25 ± 12.5	Lingual foramina	Retrospective cross-sectional in-built software (NNT) Dell Precision T5400 workstation (Dell, Round Rock, TX, USA)	Frequency and location of lingual foramina, maximum width of lingual foramina in the vertical and horizontal directions. Vertical distance of the lingual foramina to the lower border and alveolar crest of the mandible.	Quantitative
Bayrak S et al. [27]	2018	Turkey	500 (234 males, 266 females)	37.49 (range 10–87)	Lingual concavity	Retrospective cross-sectional I-CAT 3D Imaging System/I-CAT Vision software	SF types were classified as follows: type I, a flat impression < 2 mm deep; type II, a 2 to 3 mm concavity; and type III, a concavity > 3 mm. Thickness between mandibular canal and three mandibular cortical regions (inferior, buccal, lingual cortex).	Quantitative
Fouda SM et al. [28]	2020	Saudi Arabia	71 CBCT scans (26 edentulous, 45 dentate)	59.0 edentulous and 47.7 dentate	Lingual foramen	Cross-sectional Kodak 9500 Cone Beam 3D System/CS 3D	Vertical bone height, buco-lingual thickness on three different levels, presence of lingual foramen between two mandibular central incisors. The distances from the base of the lingual canal to the crest of the alveolar bone and to the base of the mandible were measured. The length of the canal was measured from the orifice of the lingual foramen to the base of the lingual canal in the same segment horizontally.	Quantitative
Genç T et al. [29]	2018	Turkey	72 (33 males, 54 females)	52.9 ± 10.7	Mandibular canal, mental foramen, lingual foramen	NA	Vertical size of the mandibular canal; distance between the mandibular canal/mental foramen and the lingual foramen and the inferior border of the mandible and alveolar crest; localization of the mental foramen—the mesial side of the first premolar—between the first premolar and second premolar, and the distal side of the second premolar; diameter and location of the lingual foramen; prevalence of the anterior alveolar loop (aAL).	Quantitative
Barbosa DA et al. [14]	2020	Brazil	646 (245 males, 401 females)	NA (six age groups)	Incisive canal	Retrospective, multicenter CS 9000 3D, Gendex CB-500, i-CAT Next Generation, i-CAT Classic/Carestream 3D	Presence or absence of the MIC. The distance from the MIC to alveolar bone crest was measured; buccal plate, lingual plate, and inferior cortex; distance from the mic to adjacent teeth apexes, MIC length, MIC vertical and horizontal diameter.	Quantitative
Alqutaibi AY et al. [30]	2022	Saudi Arabia	320 (160 males, 160 females)	41 ± 14.2	Lingual canal Lingual foramina	Retrospective study, KaVo 3D eXam, KaVo/RadiAntDICOM Viewer (Version 2022.1.1)	Presence or absence of the mandibular lingual foramen, the mandibular lingual canal, the distance from the inferior border of the mandible to the superior borer of the foramen at its lingual and buccal terminals, the distance from the buccal cortex to the beginning of the MLC, the length of the canal/canals, the diameter at the lingual and buccal terminals for each canal, the canal direction (straight or divergent/convergent).	Quantitative
Shen YW et al. [31]	2021	Taiwan	68 (30 males, 38 females)	NA	Retromolar canal—anatomical variation in mandibular canal	Asahi AZ3000/Mimics 15.0	Parameters were measured: the diameter of the retromolar foramen (RMF), the horizontal distance from the midpoint of the RMF to the distal cementoenamel junction (CEJ) of the second molar, and the vertical distance from the midpoint of the RMF to the upper border of the mandibular canal below the second molar.	Quantitative
Surathu N et al. [32]	2022	Turkey	65 mandibles (42 males and 23 females)	21 to 80 years	Lingual foramen	Cross-sectional NA	The midline of the mandible in all samples was identified using the genial tubercles. This area was then examined for lingual foramina and the distance between the superior lingual foramen and the base of the mandible in an attempt to identify whether there was a relevant anatomical consistency in the location of this foramen.	Quantitative
Sanchez-Perez A et al. [33]	2018	Spain	112 (49 males and 63 females)	50	Lingual foramen	Retrospective cross-sectional Planmeca ProMax 3D Max	Length of the exit canal; distance from the canal to the buccal, inferior, lingual cortex; distance from the canal to the closest alveolus or marginal crest; angulation of exit canal; diameter of orifice; presence of anatomical variation.	Quantitative
Vasegh Z et al. [34]	2022	Iran	38	65.16 ± 10.88 for males and 64.33 ± 10.61 for females	Buccal and lingual concavity	Cross-sectional NewTom VGi/NNT 3D software	Ridge height in anterior mandible, ridge height considering the mental foramen, ridge height in the posterior mandible considering the inferior alveolar nerve canal.	Quantitative
Gümüşok M et al. [35]	2017	Turkey	645 (281 males, 364 females)	41	Mental foramen, accessory mental foramen	Cross-section Promax 3D^®^	The numbers of AMF, its horizontal and vertical diameters, location with respect to the MF, and distance to the MF were determined. The horizontal and vertical diameters of the MF were also measured. In dentate cases, the positions of the teeth in the neighborhood of AMF were evaluated considering their root(s).	Quantitative
Wong SK et al. [36]	2018	Malaysia	100 (50 males and 50 females)	NA	Anterior loop prevalence and length	eXamVisionQ, KaVo Dental software	The IAN was traced along with the anterior loop and part of the incisive nerve, while an entry and exit of the mental foramen were analyzed from the cross-sectional view. Vertical lines were drawn on the panoramic view from the slice that corresponded to the anterior and posterior walls of the mental foramen and the most anterior point of the anterior loop length (AnLL) as seen from the cross-sectional view. The study analyzed the canal height in the region of the mental foramen and length of canal	Quantitative
Sener E. et al. [37]	2018	Turkey	70 dry mandibles (35 edentulous and 35 dentate)	NA	Mandibular incisive canal/lingual foramen	Retrospective cross-sectional/Kodak 9000 3D/Kodak Dental Imaging Software v. 6.11.6.2 and 3D module v. 2.1	Mesio-distal length of the MIC, diameters of the LF opening (lingual diameter) and lingual canal ending (labial diameter) located on the midline of the mandible and superior to the mental spine, labial and lingual heights as the distance between the lingual canal and mandibular base on the sagittal slices of the CBCT images.	Quantitative
Koivisto T et al. [38]	2016	Minnesota USA	106 (34 males and 72 females)	18 to 69 years	Mandibular canal	Retrospective cross-sectional/NA/Next Generation i-CAT	Root apices of the posterior teeth to MC with the aim to evaluate the location of the mandibular canal, bone thickness, and diameter of MC and the presence of the anterior loop.	Quantitative
Yang XW et al. [39]	2017	China	824 hemimandibles (166 males and 246 females)	11 to 81 years	Mandibular incisive canal (MIC)	Retrospective cross-sectional/NewTom VG CBCT/by QR-NNT Version 2.21 software	The anterior border of the mental foramen and the most anterior of the inferior alveolar nerve; the lengths of the anterior extension of the anterior loop (aAL) and MIC; the lengths of caudal extension of the anterior loop (cAL), the diameter of the MIC; the diameter of the MIC; the horizontal distance from the MIC to the buccal and lingual cortical borders of the mandible; and the vertical distance from the MIC to the apex of the tooth, alveolar crest, and inferior margin of the mandible.	Quantitative
Yoon TY et al. [40]	2017	Florida USA	104 (47 males and 55 females)	54.8 (21–89 years)	Posterior and anterior mandible concavity	Retrospective cross-sectional study Sirona XG3 CBCT/Galaxis/Galileo Implant Viewer	The most prominent lingual aspect of bone was marked as a reference point (SLP). A vertical line (VL) was drawn from the SLP meeting the horizontal line (HL) effectively forming the legs of a triangle. The angle of the lingual concavity (ALC) was measured using the HL of the border of the mandible that formed the hypotenuse of the triangle. The mandibular morphology type was classified as parallel, concave, and convex.	Quantitative
Zhang W et al. [41]	2015	Texas USA	59 (28 males and 31 females)	19–74 years	Alveolar ridge	Retrospective cross-sectional/ Kodak 9500 unit/Anatomage Invivo 5.1 software	All measurements were performed from molar sites in the posterior right mandible. Cross-sectional views were generated along the distal surface of the mesial root of mandibular molars or in the middle of the edentulous socket. Alveolar height, width, and buccal bone thickness were measured on the cross-sectional views separately for the first, second, and third molar.	Quantitative
J PC et al. [42]	2018	India	85	NA	Anterior loop	NA	The length of the loop was measured in mm. Five lines were drawn to standardize the nerve loop length measurement in all CBCTs.	Quantitative
Lu Cl et al. [43]	2015	California USA	122 (61 males and 61 females)	Three age groups: 21–40, 41–60, and 61–80 years	Anterior loop	Retrospective cross-sectional I-CAT Cone Beam 3D Dental Imaging System/I-CAT Vision software	Study described the precise position of a 3D image to make imaginary lines for the analysis of anterior loop length. The measure of length of the mental nerve was compared with both sided of the mandible, according to gender and sex.	Quantitative
Raju N et al. [44]	2019	Texas USA	124 (56 male and 68 females)	16–83 years	Anterior loop	Retrospective cross-sectional/ Kodak 9500 scanner/Invivo 5.0 software	Study observed the prevalence of the anterior loop, the length of the anterior loop, available bone coronally to the mental foramen and the anterior loop, passage of the anterior loop.	Quantitative
Vujanovic-Eskenazi A et al. [45]	2015	Spain	82 (26 males and 56 females)	56.56 years (range 18–80)	Mental foramen/loop	Retrospective descriptive study/ Kodak 9500 3D/NA	The distance from the lower border of the mandible to the lower point of the mental foramen and anterior extension of the mental loop to the most mesial point of the mental foramen on both types of scans.	Mixed
Kavarthapu A et al. [46]	2018	India	139 (106 males and 33 females)	15–75	Inferior alveolar nerve	Retrospective cross-sectional/NA/Sirona Galaxis Galileos Viewer Version 1.9	CN—distance from the alveolar crest to the bone directly superior to the IAN; BN—distance from the buccal cortex to the bone directly lateral to the IAN; LN—distance from the lingual cortex of the bone directly medial to the IAN; IN—distance from the inferior border of the mandible to the bone directly inferior to the IAN.	Quantitative
He X et al. [47]	2016	China	200 (97 males and 103 females)	27 (10–70)	Lingual foramina	Retrospective cross-sectional Kodak 9500 Cone Beam 3D System/CS 3D Imaging Software	Presence, frequency, location, and morphological variations. Linear measurements, as diameter, distance between the lingual foramen and the alveolar ridge crest (L1), the distance between the lingual foramen and the tooth apex (L2), and the vertical distance from the mandibular border to the lingual foramen (L3). Lingual foramina were classified as follows: in localization, occurrence, and according to the direction of the canal.	Quantitative
Çitir M et al. [48]	2021	Turkey	106 (45 males and 61 females)	55.7 ± 10.31	Lingual concavity	Retrospective cross-sectional Galileos (Sirona Dental Systems)/SIRONA Sidexis XG 2.61 viewer software	Maximum bone height and width, concavity depth, lingual slope angle, lingual concavity angle and morphology of the bone of the anterior region of the mandible were measured. Standard measures of the region 4–6 mm anterior of the mental foramen were examined. The bone in the anterior region was classified as type I lingual concavity, type II inclined to lingual, type III enlarging towards labiolingual, and type IV buccal concavity.	Mixed
Kong Zl et al. [49]	2021	China	201 (100 males and 101 females)	18–66	Lingual concavity in molar region	Retrospective cross-sectional KaVo 3D eXam/E-3D Medical Software V16.19	The long axis of the tooth determinate from the apex of the mesial root to the line that connects the midpoint of the buccolingual counterpart of the tooth crown, the long axis of the alveolar process marked by bisecting the buccal line and lingual line of the alveolar process, the angles between these imaginary lines, the width between the buccal and lingual alveolar plates at the lower point of the alveolar bone and the perpendicular line forming the angle, the long axis of the basal bone, and the upper internal angle.	Quantitative
Tsai YC et al. [50]	2021	Taiwan	149	NA	Lingual concavity (anterior)	Retrospective observational study CBCT machine NewTom 5G/software ImplantMax 4.0	The study evaluated morphological and dimensional parameters: concavity depth, concavity angle, torque, and deep bone thickness. Also, the study performed virtual implant selection and placement to define possible labial bone perforation during planning implant placement and four classes of crestal and radicular dentoalveolar phenotype.	Quantitative
Alqutaibi AY et al. [51]	2024	Saudi Arabia	150 (75 males and 75 females)	18 to 29 and over 30	Lingual concavity (posterior)	Retrospective cross-sectional study TUDH’s CBCT machine (Kavo Dental), and measurements were performed using the RadiAnt DICOM viewer	The cross-sectional view used to evaluate the morphology of the inferior mandible canal, alveolar crest, and mandibular lingual concavity in the region of the left and right first and second molar. The ridge was defined as a U, C, or P shape. Also, the most superior and inferior prominent point and deepest point of concavity were detected.	Quantitative
Kanewoff E et al. [52]	2024	Brazil	239 (98 males and 141 females)	50.9 ± 15.6	Lingual concavity	Retrospective study i-CAT CBCT system i-CAT 3D imaging system	The study performed implant placement simulation in two ways: the prosthetic-driven ideal position and bone-driven ideal position, according to the secure distance from adjacent anatomical structures. Measures performed were as follows: number and localization of lingual foramen, labial concavity angle, labial concavity depth, mandible basal bone height, tooth torque, mandibular bone thickness, angle measurement.	Quantitative
Gu L et al. [53]	2018	China	749	37.5 ± 13.6	Mandibular canal	Retrospective CBCT scanner (Planmeca, Planmeca Romexis software)	Position of the mandibular canal relative to the roots in four class (apical, buccal, lingual, and interradicular) and contact relation of the mandibular third molar and the mandibular canal in each class.	Qualitative

**Table 2 diagnostics-15-00155-t002:** Bias assessment.

Author of the Study	D1	D2	D3	D4	D5	D6	D7	Overall
Aljarbou FA et al. 2019 [25]								☺
Sekerci AE et al. 2014 [26]								☺
Bayrak S et al. 2018 [27]								☺
Fouda SM et al. 2020 [28]								☺
Genç T et al. 2018 [29]								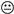
Barbosa DA et al. 2020 [14]								☺
Alqutaibi AY et al. 2022 [30]								☺
Shen YW et al. 2021 [31]								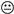
Surathu N et al. 2022 [32]								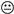
Sanchez-Perez A et al. 2018 [33]								☺
Vasegh Z et al. 2022 [34]								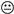
Gümüşok M et al. 2017 [35]								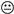
Wong SK et al. 2018 [36]								☺
Sener E. et al. 2018 [37]								☺
Koivisto T et al. 2016 [38]								☺
Yang XW et al 2017 [39]								☺
Yoon TY et al. 2017 [40]								☺
Zhang W et al. 2015 [41]								☺
J PC et al. 2018 [42]								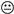
Lu Cl et al. 2015 [43]								☺
Raju N et al. 2019 [44]								☺
Vujanovic-Eskenazi A et al. 2015 [45]								☺
Kavarthapu A et al. 2018 [46]								☺
He X et al. 2016 [47]								☺
Çitir M et al. 2021 [48]								☺
Kong Zl et al. 2021 [49]								☺
Tsai YC et al. 2021 [50]								☺
Alqutaibi AY et al. 2024 [51]								☺
Kanewoff E et al. 2024 [52]								☺
Gu L et al. 2018 [53]								☺

D1: risk of bias due to confounding, D2: risk of bias in classification of interventions, D3: risk of bias in selection of participants into the study (or into the analysis), D4: risk of bias due to deviations from intended interventions, D5: risk of bias due to missing data, D6: risk of bias arising from measurement of the outcome, D7: risk of bias in selection of the reported result. Green color indicates studies with little or no risk of bias for specific domains, gray boxes indicate studies where a specific domain was not examined, while red boxes indicate studies with a higher and high risk of bias. The last column, with symbols, indicates the overall risk of bias assessment for the entire study: green indicates a low level of bias (☺), yellow indicates a concern (
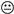
), and red indicates a high level of bias (☹).

**Table 3 diagnostics-15-00155-t003:** Overall mean and standard deviation values of the most common parameters considered for the inferior alveolar canal, presented in millimeters.

Author	Distance to Buccal Cortical Plate	Distance to Lingual Cortical Plate	Distance to Alveolar Crest	Distance to Inferior Border of Mandible	Vertical Size of Canal	Shortest Distance from Canal to Tooth Apices
Aljarbou et al. [25]	4.64 ± 1.63	3.63 ± 1.40	NA	NA	NA	3.16 ± 1.95
Genç et al. [29]	NA	NA	11.58 ± 3.42	7.58 ± 1.33	2.53 ± 0.51	NA
Kavarthapu et al. [46]	4.68 ± 1.34	2.48 ± 1.15	15.04 ± 3.32	7.23 ± 1.73	NA	NA

**Table 4 diagnostics-15-00155-t004:** Overall mean and standard deviation values of the most common parameters considered for the mental foramen and anterior loop, presented in millimeters.

Author	MF and Alveolar Crest	MF and Base of Mandible	MF Vertical Diameter	MF Horizontal Diameter	AL Length	AL and Alveolar Bone	AL and Base of Mandible
Genç et al. [29]	9.23 ± 3.26	11.50 ± 1.86	NA	NA	NA	NA	NA
Gümüşok et al. [35]	NA	NA	3.11 ± 0.89	2.80 ± 0.99	NA	NA	NA
Wong et al. [36]	NA	NA	NA	NA	3.77 ± 1.74	NA	NA
J, PC et al. [42]	NA	NA	NA	NA	2.79 ± NA	NA	NA
Lu et al. [43]	NA	NA	NA	NA	1.46 ± 1.25	NA	NA
Raju et al. [44]	12.92 ± 3.55	NA	NA	NA	1.73 ± NA	17.06 ±2.69	NA
Vujanovic-Eskenazi et al. [45]	NA	NA	NA	NA	1.59 ± 0.93	NA	11.43 ± 1.81

**Table 5 diagnostics-15-00155-t005:** Overall mean and standard deviation values of the most common parameters considered for lingual foramina and the lingual canal, presented in millimeters.

	Distance from Alveolar Crest	Distance from Base of Mandible	Distance from Buccal Cortex	Distance from Lingual Cortex	Length of Canal	Diameter of Canal
Sekerci et al. [26]	20.30 ± 4.59	9.31 ± 3.86	NA	NA	NA	1.02 ± 0.37
Fouda et al. [28]	17.99 ± 4.86	8.05 ± 2.37	NA	NA	8.30 ± 1.99	NA
Genc et al. [29]	13.62 ± 4.35	11.35 ± 3.41	NA	NA	NA	NA
Alqutaib et al. [30]	NA	NA	NA	NA	5.49 ± 1.89	NA
Surathu et al. [32]	NA	15.06 ± 1.91	NA	NA	NA	NA
Sanchez-Perez et al. [33]	11.80 ± 3.25	7.75 ± 2.51	4.94 ± 1.60	8.62 ± 1.36	8.93 ± 1.94	1.44 ± 0.45
Sener et al. [37]	NA	NA	NA	NA	NA	0.78 ± 0.30
He et al. [47]	16.62 ± 8.57	13.66 ± 8.51	NA	NA	NA	0.86 ± 0.32

**Table 6 diagnostics-15-00155-t006:** Overall mean and standard deviation values of the most common parameters considered for the mandibular incisive canal, presented in millimeters.

Author	MIC to Alveolar Crest—Initial	MIC to Buccal Cortex—Initial	MIC to Lingual Cortex—Initial	MIC to Base of Mandible—Initial	MIC to Alveolar Crest—End	MIC to Buccal Cortex—End	MIC to Lingual Cortex—End	MIC to Base of Mandible—End	Incisive Canal Length	Incisive Canal Diameter
Barbosa et al. [14]	16.48 ± 5.52	2.62 ± 1.26	5.12 ± 1.71	9.40 ± 1.88	18.72 ± 3.47	4.00 ± 1.41	4.59 ± 1.66	8.87 ± 2.01	NA	NA
Sener et al. [37]	NA	NA	NA	NA	NA	NA	NA	NA	2.82 ± 1.26	2.40 ± 0.65
Yang et al. [39]	NA	NA	NA	NA	NA	NA	NA	NA	10.50 ± 4.81	2.01 ± 0.47

**Table 7 diagnostics-15-00155-t007:** Overall mean and standard deviation values of the most common parameters considered for mandibular concavity, presented in millimeters.

Author	Anterior Concavity Depth	Anterior Concavity Length	Anterior Angulation	Posterior Concavity Depth	Posterior Concavity Length	Posterior Angulation
Bayrak et al. [27]	NA	NA	NA	2.69 ± 0.84	NA	NA
Vasegh et al. [34]	1.47 ± 0.58	NA	10.20 ± 5.34	1.85 ± 0.52	NA	10.66 ± 5.96
Yoon et al. [40]	2.54 ± 1.08	18.23 ± 3.55	83.27 ± 8.68	3.84 ± 1.98	16.10 ± 2.89	75.48 ± 12.71
Çitir et al. [48]	3.66 ± 0.97	NA	60.78 ± 2.70	NA	NA	NA
Tsai et al. [50]	4.44 ± 1.96	NA	NA	NA	NA	NA
Alqutaibi et al. [51]	NA	NA	NA	1.98 ± 0.78	11.73 ± 2.48	47.95 ± 11.83
Kanewoff et al. [52]	3.20 ± 1.50	NA	NA	NA	NA	NA

## Data Availability

Data available on request.

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
