# Peer review of "Anatomic Variations Important for Dental Implantation in the Mandible—A Systematic Review"

_diagnostics, 2025, doi:10.3390/diagnostics15020155_

Round 1

Reviewer 1 Report

Comments and Suggestions for Authors

This is a systematic review on the topic of anatomical landmarks and variations in the mandible that influence dental implants placement.

Congratulations to the authors on your work on a topic of interest to oral surgeons.

The article is well written, easily readable. The materials and methods used in this systematic review are clearly presented.

The results read well thanks to the included tables.

My comments are as follows:

The Abstract seems a bit vague, the results and conclusions should be given more in detail

The Conclusion should be somewhat expanded as well as future lines of research 

Check the style of the references, the names of the papers are not always uniform.

Author Response

Dear reviewer thank you for your kind words and valuable comments. We did change everything that you suggested. Thank you for great feedback.

Comment 1: The Abstract seems a bit vague, the results and conclusions should be given in more detail

Response: The result and conclusion are rewritten in the Abstract. Thank you for the suggestion.

Comment 2: The Conclusion should be somewhat expanded as well as future lines of research 

Response: The conclusion was expanded as suggested.

Comment 3: Check the style of the references, the names of the papers are not always uniform.

Response: It is checked and rewritten where mistakes were. Thank you for the detailed check.

Sincerely,

Reviewer 2 Report

Comments and Suggestions for Authors

Although the content is highly novel and interesting as a systematic review theme, the results and tables are poorly written, and the paper is disappointing. However, if you change the way the tables and results are written, I think it will become a great systematic review that will be cited by others, so please do so.   However, with regard to the results, the paper only includes summaries by each author, and the overall average is not calculated, so it is similar to a simple narrative review. Please include the overall average, etc., in the abstract and as a systematic review.   

1) There is no summary of the results for Table 1. As this is a systematic review, please include a summary of the results for each paper. Also, I think it would be better to include a table with the results for each item in Table 1, summarising the average for each.  

2) The way the results are described is not a systematic review, but rather the way a narrative review is written, as it only describes the results for each author. This is the biggest problem with this paper. Rather than just describing the results of each author for each item (Inferior alveolar canal (IAC), Mental foramen (MF) and anterior loop (AL), Lingual foramina/lingual canal (LF/LC), Mandibular incisive canal (MIC), Mandibular concavity), you should write the average, gender differences, and evaluation methods as a systematic review. This also applies to the abstract.

Simply selecting papers is not a systematic review. I think a systematic review is one that evaluates the results that can be derived from the selected papers.  

Author Response

We want to thank the reviewer for taking the time to read and give valuable comments and suggestions for improving our work.  Our answers are following your questions/suggestions.  

Comments 1: There is no summary of the results for Table 1. As this is a systematic review, please include a summary of the results for each paper. Also, I think it would be better to include a table with the results for each item in Table 1, summarising the average for each. 

Response: A summary of the results is presented for each study included in this review article, and they have been written in detail in subsections named after observed anatomical landmarks (mental foramen, anterior loop, lingual foramina/lingual canal, mandibular incisive canal and mandibular concavity), the characteristics of the articles are presented in Table 1 using SPIDER method. As already mentioned, in the limitation of the study, all named anatomical characteristics and variations did not always measure from the same anatomical points and did not consider the same groups, so presenting the overall mean was challenging. But also, this was the reason why we have chosen to deal with this subject and separated the ways of how certain parameters were measured in research articles, in summarizing Table, in the period of ten years. We wanted to highlight the variability of measured anatomical points because measuring them on different levels can be an obstacle for future researchers and clinicians to choose which point should be used in their evaluations. We thank you for your great suggestion and we presented the individual mean values of measured anatomical details in the separated tables, for the measured points that were the most common among certain anatomic landmarks. Tables can be found below each subsection of results from Table 3 to Table 7. We also expanded the explanation of this issue in the limitations of the study.

Comments 2: The way the results are described is not a systematic review, but rather the way a narrative review is written, as it only describes the results for each author. This is the biggest problem with this paper. Rather than just describing the results of each author for each item (Inferior alveolar canal (IAC), Mental foramen (MF) and anterior loop (AL), Lingual foramina/lingual canal (LF/LC), Mandibular incisive canal (MIC), Mandibular concavity), you should write the average, gender differences, and evaluation methods as a systematic review. This also applies to the abstract. Simply selecting papers is not a systematic review. I think a systematic review is one that evaluates the results that can be derived from the selected papers.

Response: Systematic articles can be classified as ones with qualitative and quantitative synthesis, or both. The selection of the articles was not as simple, as may perceived, their selection was in many different ways, as described throughout the text. After the initial search of articles based on the detailed PRISMA suggestions, the process of their selection is presented first in the flow chart. All articles were read in detail; they had to pass the assessment process that is described in the text – the reliability and afterward critical bias assessment (done by the latest update in 2024) that was also presented through the text and presented in Table 2. This research followed each PRISMA statement step by step. The characteristics of studies are presented in Table 1 by the specific SPIDER approach. A narrative review does not contain the same methodological approach and it is much easier to execute, providing fewer details. We thank you for your valuable comments and we are adding the new tables with the overall mean, as you suggested, for the most commonly measured points for certain landmarks (Table 3 to Table 7) while differences in measured details and statistical significance between age, gender, side of the mandible, teeth present are presented in the text of each subsection, above the mentioned tables.  

Sincerely,

Round 2

Reviewer 2 Report

Comments and Suggestions for Authors

It has been appropriately revised and contains clinically useful information.

The content is acceptable.

Please also attach the reference numbers to Tables 3 to 7.

Author Response

Dear Reviewer, 

Thank you for your feedback and comments. It was very helpful and the article looks much better. 

Comments: Please also attach the reference numbers to Tables 3 to 7.

Response: References are attached to Tables 3 to 7.

Sincerely,
